# An Urban Image Stimulus Set Generated from Social Media

Ardaman Kaur [1], André Leite Rodrigues [2], Sarah Hoogstraten [1], Diego Andrés Blanco-Mora [3], Bruno Miranda [3,4], Paulo Morgado [5] and Dar Meshi [1,*]

[1] Department of Advertising and Public Relations, Michigan State University, East Lansing, MI 48824, USA; kaurard1@msu.edu (A.K.); hoogstr9@msu.edu (S.H.)

[2] Centre of Geographical Studies, Institute of Geography and Spatial Planning, University of Lisbon, 1600-276 Lisbon, Portugal; andrerodrigues1@edu.ulisboa.pt

[3] Institute of Physiology, Lisbon School of Medicine, University of Lisbon, 1649-004 Lisbon, Portugal; diegoandres@edu.ulisboa.pt (D.A.B.-M.); bruno.miranda@campus.ul.pt (B.M.)

[4] Institute of Molecular Medicine, University of Lisbon, 1649-004 Lisbon, Portugal

[5] Centre of Geographical Studies, Associate Laboratory TERRA, Institute of Geography and Spatial Planning, University of Lisbon, 1600-276 Lisbon, Portugal; paulo@campus.ul.pt

* Correspondence: darmeshi@msu.edu

**Abstract:** Social media data, such as photos and status posts, can be tagged with location information (geotagging). This geotagged information can be used for urban spatial analysis to explore neighborhood characteristics or mobility patterns. With increasing rural-to-urban migration, there is a need for comprehensive data capturing the complexity of urban settings and their influence on human experiences. Here, we share an urban image stimulus set from the city of Lisbon that researchers can use in their experiments. The stimulus set consists of 160 geotagged urban space photographs extracted from the Flickr social media platform. We divided the city into 100 × 100 m cells to calculate the cell image density (number of images in each cell) and the cell green index (Normalized Difference Vegetation Index of each cell) and assigned these values to each geotagged image. In addition, we also computed the popularity of each image (normalized views on the social network). We also categorized these images into two putative groups by photographer status (residents and tourists), with 80 images belonging to each group. With the rise in data-driven decisions in urban planning, this stimulus set helps explore human–urban environment interaction patterns, especially if complemented with survey/neuroimaging measures or machine-learning analyses.

**Dataset:** The urban image stimulus set is available for download on the Open Science Framework (OSF) (https://osf.io/c79w5/, accessed on 23 November 2023).

**Dataset License:** The urban image stimulus set is being made available under the CC0 1.0 Universal license.

**Keywords:** social media; urban photographs; human–urban environment interactions

## 1. Summary

Social media have become an integral part of people's everyday lives. More than four billion individuals use social media globally, and it is anticipated that this number will grow to almost six billion by 2027 [1]. The widespread use and influence of social media have drawn the interest of researchers from diverse areas, leading to an increase in scientific studies [2–4]. Social media platforms like Facebook, Instagram, X/Twitter, and TikTok provide rich and valuable data sources in the form of text, photos, videos, likes, comments, shares, and other interactions [5]. Some visual media-sharing platforms, like Flickr, also offer the possibility for users to upload, store, and share photo and video data with other users [6,7]. Moreover, certain platforms (Facebook, X/Twitter, etc.) also allow users to add geographical location tags to their posted content to indicate the user's location and/or

where the content was captured. Since our usage and dependency on mobile devices continue to rise, people have become a crucial source for collecting, retrieving, sharing, and disseminating various types of information.

Geotagged social media data also form a significant aspect of volunteered geographic information (VGI), which is a concept that depends on the voluntary contributions of individuals who are willing to share location-based information through digital platforms [8–10]. Within VGI, there is a subset that falls in the category of citizen science, involving non-professional scientists or citizens in data collection and, to some extent, analysis [11]. Social media users also serve as passive contributors to citizen science initiatives (a subset of VGI) when they share geotagged content that has the potential to be utilized in scientific research [12]. Crowdsourcing geotagged social media data can be a valuable technical and methodological tool for researchers and analysts in various fields, including environmental studies, psychology, transport and urban planning, tourism, behavioral economics, and other social sciences [7,11,13,14].

Geotagged social media content plotted on maps allows for visualizing the distribution and concentration of these data across different locations, giving insights into preferences (most sought-after locations), popular activities, interests, and emerging trends [13,15,16]. Furthermore, the number of views and "likes" that these geotagged media receive often indicates their popularity on social media, which can help researchers understand both online and offline social behavior [17]. For instance, Tenkanen et al. [18] compared the monthly official visitor statistics in 56 recreational protected areas (national parks) in 2014 to the visitation metric derived from Flickr, Instagram, and Twitter (currently known as X). The official visitor statistics comprised monthly data from installed electronic counters and sold entrance tickets, and the number of social media users posting content from these national parks aggregated for each month was the measure of social media-derived visitation statistics. The results showed a consistent relationship between monthly official visitor counts and social media-derived visitation statistics across all three platforms combined. Thus, geotagged social media data can serve as a valuable source of information for monitoring visitor numbers and gaining insights into a place's popularity and visiting patterns over time.

It is important to highlight that global urbanization has been rapidly increasing, with 57% of the world's population currently living in urban areas [19], and, according to the United Nations, this percentage will reach 68% by 2050 [20]. Consequently, numerous studies have emerged that link the well-being of individuals to the urban environment that they live in [21–23]. A recent study spanning 60 developed countries and including 230 million people established a positive link between urban green space and a nation's happiness level [24]. This study's urban green space amount was measured using the Normalized Difference Vegetation Index (NDVI) computed from high-resolution satellite images for different countries. Geotagged data can also be employed at a regional and community level by analyzing neighborhood characteristics such as demographics and environmental factors to improve the quality of life for individuals. Another study by Stier and colleagues [25] used geotagged Twitter (currently known as X) datasets to identify words related to depressive symptoms in users' tweets. They found that larger urban areas in the US with denser socioeconomic network connections had lower rates of depression. Geotagged data can thus provide vital insights into the impact of the urban environment on mental health and well-being [26]. In sum, geotagged data can play an important role in urban planning and development initiatives, enabling planners to make data-driven decisions to improve urban livability and sustainability and help policymakers to derive policy recommendations tailored to local needs.

To facilitate research in the domains mentioned above, we created a rich stimulus set of 160 urban space images of Lisbon. We sourced these images from the Flickr social media platform, and all images were geotagged, with a linked owner identification tag and upload date. We divided the city into $100 \times 100$ m cells to calculate the cell image density (number of images in each cell) and cell green index (Normalized Difference Vegetation

Index for each cell). Then, we assigned these values to each image based on their geotagged information. In addition, we also provide the popularity of each image (normalized views on the social network). Finally, we categorized these images into two putative groups by photographer status (residents and tourists), with 80 images belonging to each group.

In addition to our stimulus set, other researchers have created valuable datasets that include urban space images, such as Placepulse [27], Cityscapes [28], and ADE20K [29]. Of note, Placepulse also contains geotagged images from Lisbon, similar to our stimulus set. However, this dataset does not explicitly provide several relevant metrics we computed, such as popularity or the amount of greenery associated with each image. Instead, it offers pairwise comparisons and rankings for each image based on attributes like safety, liveliness, and aesthetics. ADE20K and Cityscapes, on the other hand, do not have geotagged images, nor do they provide other specific metrics (for greenery or popularity of the image). We believe that our stimulus set (used alone or possibly in conjunction with other available urban space datasets) aligns with the collaborative nature of VGI and could provide a strong foundation for future studies, especially for exploring urban environment–human interaction patterns. This stimulus set can be a valuable tool for urban planners to better understand how people move across space through time and what sort of activities they perform through behavioral and neurocognitive measures, as well as for social media researchers to assess various aspects of online social behavior.

## 2. Data Description

We have provided this urban image stimulus set for download on the Open Science Framework (OSF) (https://osf.io/c79w5/, accessed on 23 November 2023). The stimulus set images are stored in the folder labeled "Urban image stimulus set" and are numbered from 1–160. The data also include an Excel sheet named "Urban image stimulus set variables" with columns that provide information on the variables associated with each image, namely the image number, owner tag, photo ID (PID), secret tag, category (presumed residents/tourists), bin number (one to eight, representing different ranges of the cell image density), cell image density, cell green index, number of views, normalized popularity, brightness, and contrast.

## 3. Methods

We developed our image stimulus set in a four-step process. The first step consisted of extracting bulk geotagged urban environment images from the Flickr social media platform (image extraction process). The second step involved determining four variables associated with the images extracted in the first step (image variable determination). The third step consisted of selecting images that depicted urban spaces with respect to our four variables (image selection process). The fourth and final step involved assessment of the brightness and contrast of the images selected in the third step (image brightness and contrast assessment). We adjusted any images that had brightness and contrast values beyond three standard deviations and did not discard any images during this step. We describe each of these steps below, in greater detail.

### 3.1. Step 1: Image Extraction Process

We obtained images from the city of Lisbon (Portugal) that were posted to the social media platform Flickr. To do this, we used the Flickr API and created a Python query to search for geotagged images with respect to the date they were uploaded and the specific location where the image was taken. Figure 1 depicts these input parameters, with the date duration between 1 January 2016 and 29 September 2021 and the bounding box covering the boundaries of the Lisbon municipality [search coordinates in WGS84: $-9.23$, 38.69, $-9.09$, 38.80; search area: 158.43 km$^2$]. The above process resulted in 75,233 images taken in the city of Lisbon. We retrieved the geotagged location, the information about the photographer (owner id), and details related to the Flickr account (highest number of views, etc.); such data extraction was the basis to determine the variables explained in the subsequent step.



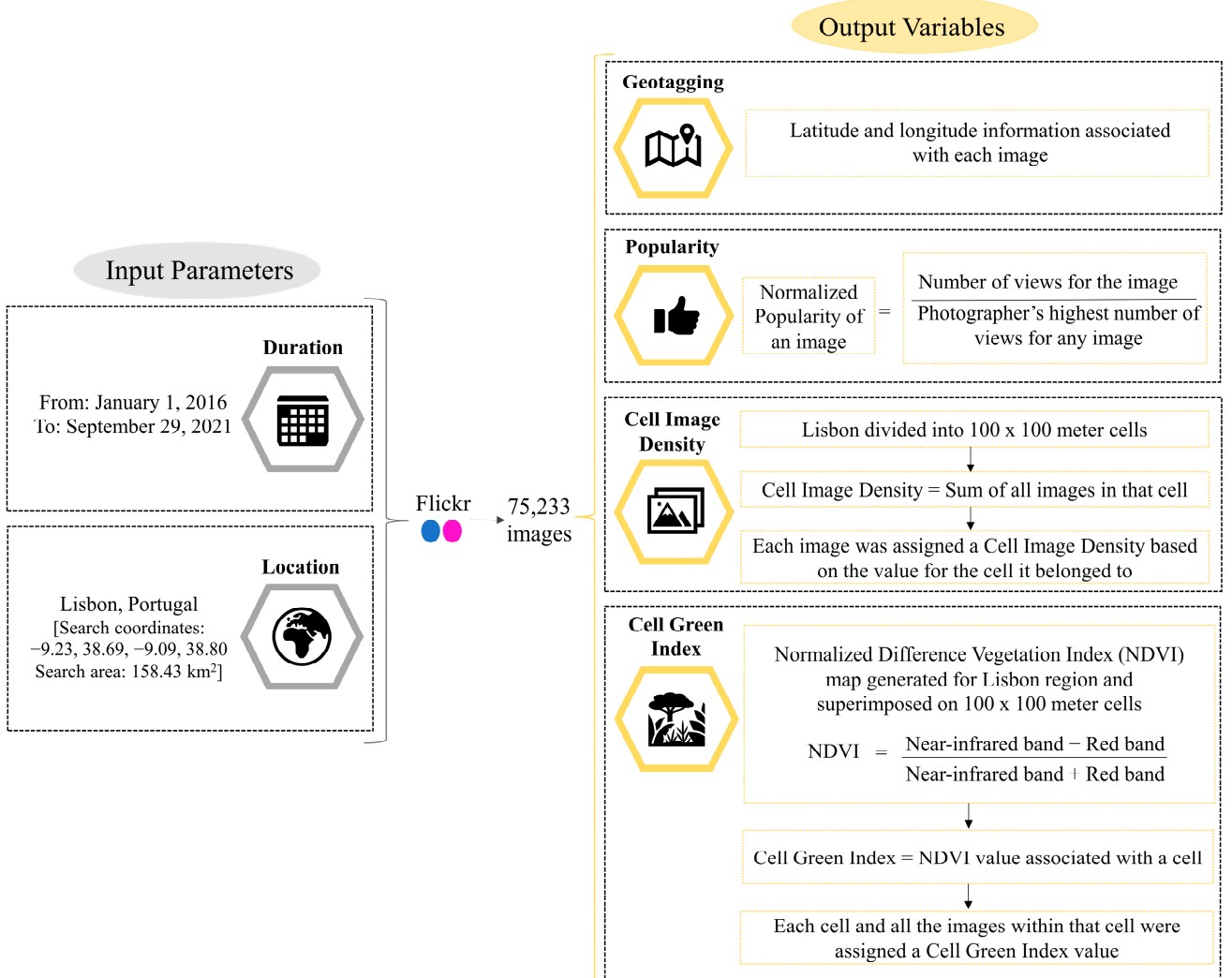

**Figure 1.** Process for extracting 75,233 urban environment images from the Flickr website. We selected images based on the input parameters (**left**) and determined the output variables (**right**) for all images.

### 3.2. Step 2: Image Variable Determination

In this step, we determined the following variables for each of the extracted images (see Figure 1).

### 3.2.1. Geotagging

All images extracted in Step 1 were geotagged. We, therefore, fetched each image's associated latitude and longitude coordinates.

### 3.2.2. Normalized Popularity

We calculated this variable by dividing the number of views for each image by the highest number of views for any image that the photographer of the image had posted. In this way, we created a normalized index score for the popularity of each image that ranges from 0 to 1. For example, if the image in question had been seen by 100 people, and the photographer's most popular image had been seen by 1000 people, then 100/1000 = 0.1 normalized popularity score.

### 3.2.3. Cell Image Density

We divided the entire 158.43 km$^2$ search area of Lisbon into $100 \times 100$ m cells using the Fishnet method in ArcGIS 10.7. Using each image's geotag data, we calculated the total number of extracted images in each $100 \times 100$ m cell included within the geographical boundaries of the city of Lisbon. This provided a cell image density specific to each cell—which reflects the total number of geotagged photographs shared by Flickr users in the corresponding area of the cell during the specified time period. We then associated this cell image density with each image that was taken within that cell.

### 3.2.4. Cell Green Index

We quantified the overall greenness of the cell by using the Normalized Difference Vegetation Index (NDVI) [30]. NDVI measures the ratio of the difference in the red and near-infrared portions of the spectrum to their respective sum. Healthy vegetation (chlorophyll) reflects more near-infrared light than other wavelengths, but it absorbs more red light. Thus, NDVI ranges from $-1.0$ to $1.0$, with larger positive values indicating green vegetation. Non-vegetated areas, including bare soil, open water, snow/ice, and most construction materials, have much lower NDVI values [31]. The NDVI is preferred to the simple index for global vegetation monitoring because it compensates for changing illumination conditions, shadows, surface slope, and aspect, among other factors. NDVI has achieved good results in detecting green cover, monitoring land surfaces and vegetation canopies, estimating leaf area index, estimating grass cover vegetation biomass, and quantifying the percentage of grass cover [32]. We employed satellite imagery from the Sentinel-2 data source and created an NDVI map for the city of Lisbon in ArcGIS. We ensured accuracy by creating a synthesis map that averages all months for each year (2016–2021). This provided a comprehensive overview of vegetation changes in Lisbon over a six-year period. For instance, if an image taken in 2016 shows an empty spot, but a building is constructed on that same spot by 2020, using the NDVI from 2020 would not give an accurate reflection of the vegetation from 2016. We overlaid this map onto a grid of $100 \times 100$ m cells, and each cell and image within it received the same green index value for the year it was photographed.

### 3.3. Step 3: Image Selection Process

Past research has focused on understanding the dynamics of spatial interactions between residents and tourists and their implications on urban planning [33,34]. Thus, to expand the scope of research that can be performed utilizing these images, we divided the images into two categories: images photographed by residents and images photographed by tourists. The following paragraphs describe our approach in performing this segregation and the successive steps to select the relevant images depicting the urban spaces of Lisbon.

We segregated the 75,233 images obtained in Step 1 into two categories (presumed residents or tourists) based on data from the photographer's Flickr account (see Figure 2). We assigned an image to the presumed resident category if the photographer's Flickr account uploaded images within our defined geographical boundaries in the city of Lisbon for more than three consecutive months. Conversely, we assigned an image to the presumed tourist category if the photographer's Flickr account uploaded images for less than three consecutive months. Of note, according to data published by the National Institute of Statistics of Portugal (NISP) in 2020 [35], tourists stay an average of around three nights in Portugal. Importantly, NISP provide this average stay datapoint without a standard deviation value (s.d.) to calculate a threshold that could help us improve our residents vs. tourists categorization (e.g., average + 3 s.d.). Thus, given the missing s.d., we adopted a conservative categorization threshold of three months. To note, however, we acknowledge that some residents' photographs will likely be labeled as tourists' (and vice versa). In support of this, we have termed this categorization "presumed" residents and tourists throughout this manuscript. In addition, it is important to clarify that we only considered the number of unique days on which a Flickr user posted images, rather than the total quantity of images they posted. Please see Figure 2 for a depiction of the cell image density

distribution for the 75,233 images, as well as the images we categorized as photographed by presumed residents (26,585 images) and presumed tourists (48,648 images).

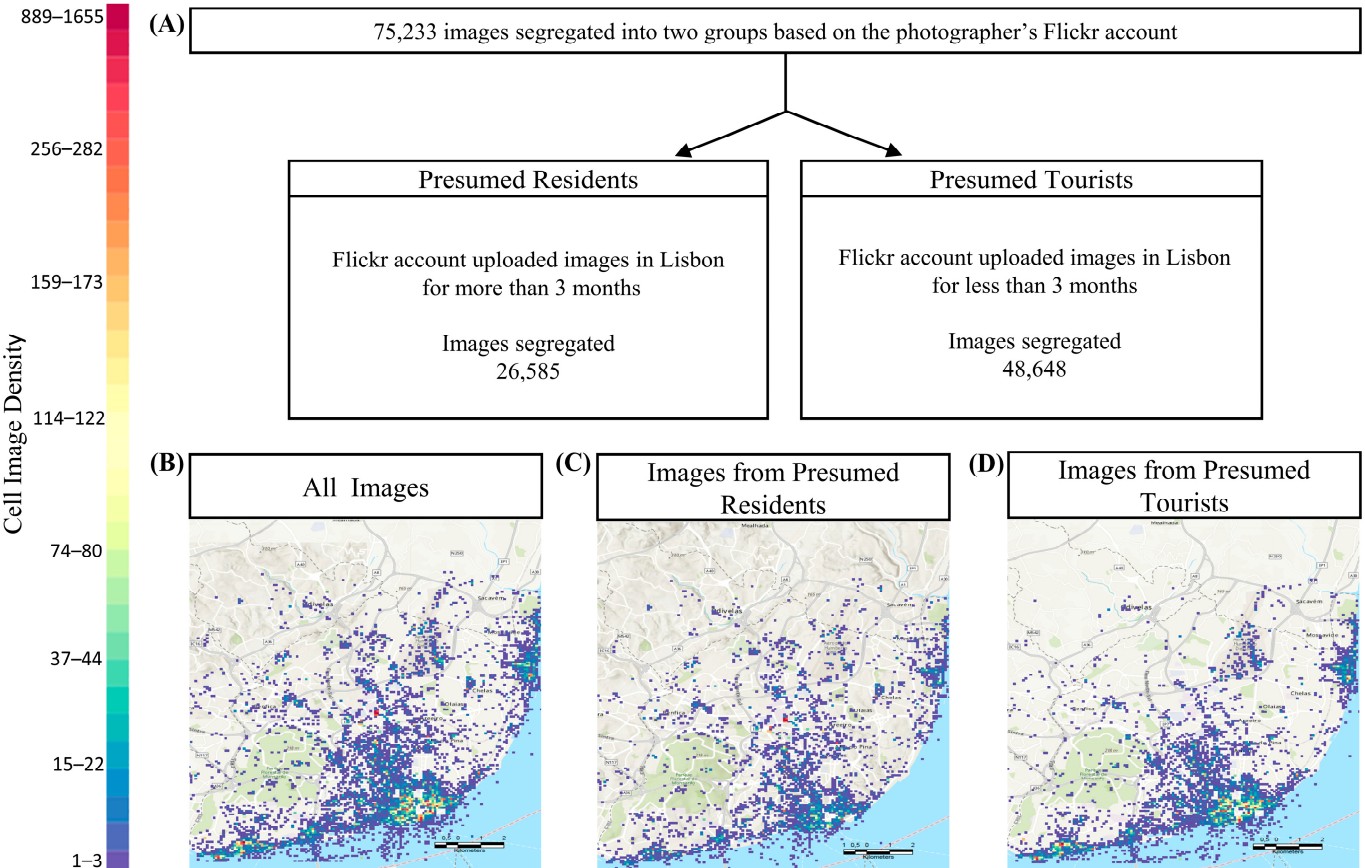

**Figure 2.** (**A**) Process for segregating 75,233 images into two groups: photographs uploaded by presumed residents or photographs uploaded by presumed tourists; (**B**) cell image density distribution in each 100 × 100 m cell for the entire set of 75,233 images; (**C**) cell image density distribution for images uploaded by presumed residents; (**D**) cell image density distribution for images uploaded by presumed tourists. The color bar denotes the cell image density, where blue represents the minimum values, and red represents the maximum values.

We next took further steps to refine our stimulus set by independently handling images belonging to the presumed resident and tourist categories. All the analyses were conducted using MATLAB R2022a. Firstly, we divided the images in each category into eight bins based on cell image density (Figure 3A). Next, we aimed to select 10 relevant images depicting Lisbon's urban spaces for each bin in each category, which would result in 160 total images (80 images for tourists and 80 images for residents). Importantly, we intended to select an image set where our key image variables (popularity, cell image density, and cell green index) were orthogonal to each other. We did this to help researchers avoid collinearity effects when conducting future investigations with these image variables. We achieved this with an iterative image selection process involving first plotting all images in a bin and then subjecting individual images within the bin to an in-house algorithm (Figure 3B). To explain in more detail, we first created a bivariate histogram with all images in a single bin plotted according to their popularity and cell green index. Second, we divided the histogram into three equal parts along the cell green index axis. We decided to divide this axis into thirds of its original divisions after analyzing the histogram's distribution across all bins in both categories. For example, if the original divisions in the histogram along the cell green index for a particular bin were 33, they were divided into

three parts, 11 divisions each, for further steps. To note, before proceeding with this three-part division, there were a small number of bins where the number of original divisions along this axis was even, so we replotted the histogram by modifying the original division number to the following odd number. Next, we randomly selected one image from each part, obtaining three images in one complete iteration. We then ran four iterations of this selection process to retrieve 12 distributed images from the bin we were working on. At this step, we visually inspected the 12 images to confirm that they depicted urban spaces in Lisbon. If some images were visually not appropriate (e.g., a close-up image of a face), we removed them and, if needed, ran more iterations until we obtained 10 visually relevant images for each bin. In a few bins with a high percentage of visually irrelevant images, we had to manually select some images after our algorithm reached saturation (repeatedly selecting irrelevant images). In these bins, our algorithm selected ~70% of the final images. We stopped the process when we obtained 10 images for each bin.

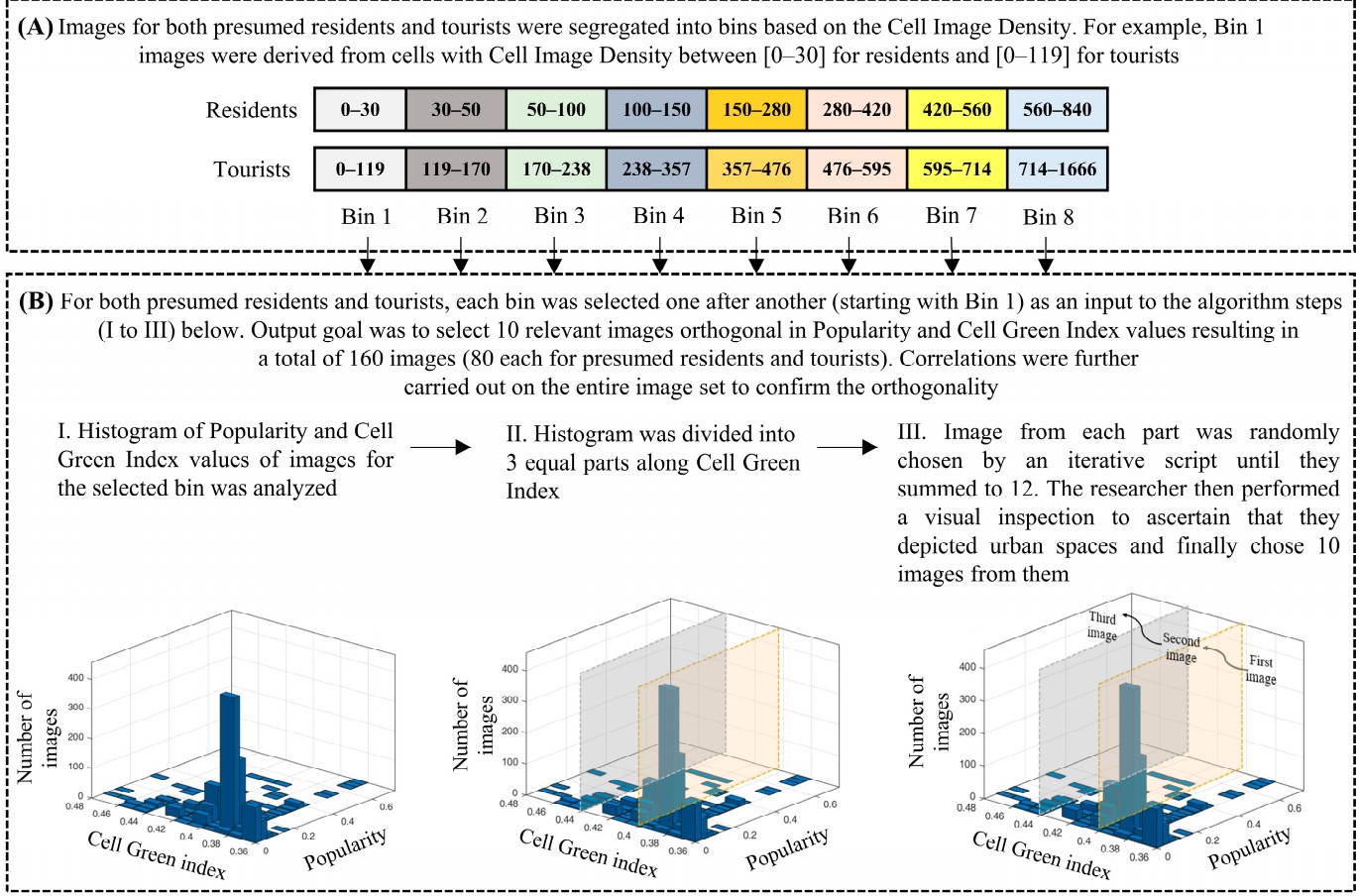

**Figure 3.** Process for selecting 160 urban environment images (80 each from the presumed resident and tourist categories). (**A**) Segregating images based on cell image density into bins for both presumed residents and tourists (**B**) Selecting each bin sequentially and feeding them to an in-house algorithm to select 10 images per bin orthogonal in popularity and cell green index. This process allowed for images distributed across our image cell density variable and assured that images were not correlated with respect to our selected variables (popularity and cell green index).

To validate our selection process and confirm orthogonality between variables, we used a MATLAB-based robust correlation toolbox [36] to conduct three different types of correlations (Pearson, percentage bend, and Spearman) on the entire set of 160 images (Figure 4). We also conducted these correlations on the presumed residents and tourists category sets of 80 images each (Table 1). All correlations were weak and non-significant

($p > 0.05$), confirming the orthogonality between the output variables of popularity, cell image density, and cell green index for the entire stimulus set (and also for each residents and tourists category sets of images).

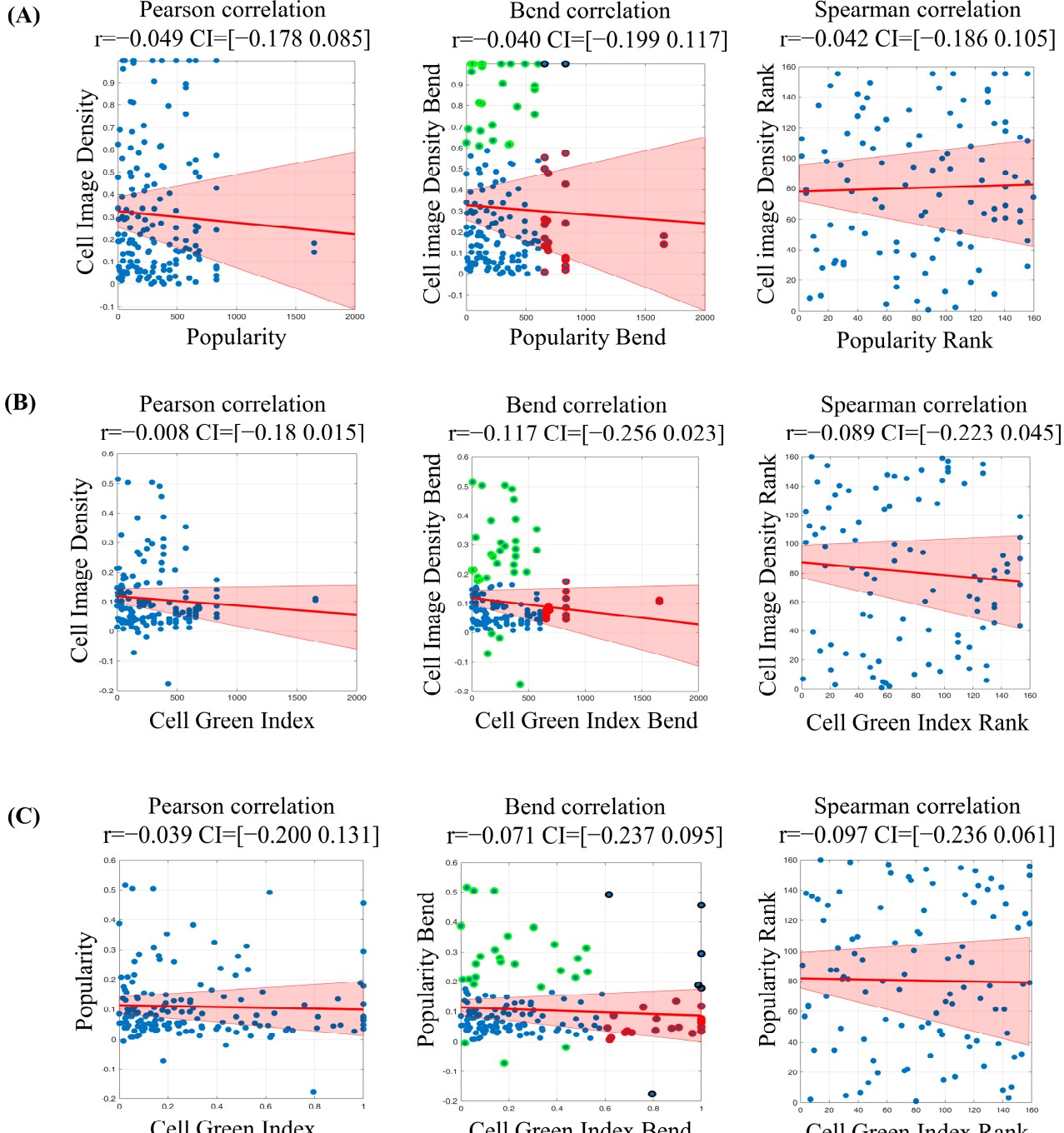

**Figure 4.** Pearson, Bend, and Spearman correlation plots with the 95% bootstrapped confidence intervals (shaded areas) for the entire set of 160 urban environment images. (**A**) Correlations between cell image density and popularity, (**B**) correlations between cell image density and cell green index, and (**C**) correlations between popularity and cell green index. For the bend correlation, red indicates data bent in the variable on the x-axis, green in the variable on the y-axis, and black indicates data bent in both variables.

**Table 1.** Pearson, Bend, and Spearman correlation values with significance levels for correlation among cell image density, popularity, and cell green index performed separately for the images uploaded by presumed residents and tourists. For the identical analyses performed on the full set of images, please see Figure 4.

| Category | Correlation Variables | Pearson Correlation | | Bend Correlation | | Spearman Correlation | |
|---|---|---|---|---|---|---|---|
| | | r | p | r | p | r | p |
| Images by Presumed Residents | Cell Image Density and Popularity | −0.069 | 0.546 | −0.061 | 0.588 | −0.039 | 0.731 |
| | Cell Image Density and Cell Green Index | −0.122 | 0.281 | −0.195 | 0.082 | −0.212 | 0.06 |
| | Popularity and Cell Green Index | −0.037 | 0.744 | −0.098 | 0.387 | −0.121 | 0.284 |
| Images by Presumed Tourists | Cell Image Density and Popularity | −0.136 | 0.228 | −0.149 | 0.188 | −0.175 | 0.120 |
| | Cell Image Density and Cell Green Index | −0.092 | 0.419 | −0.107 | 0.345 | −0.040 | 0.724 |
| | Popularity and Cell Green Index | −0.059 | 0.602 | −0.096 | 0.394 | −0.103 | 0.363 |

*3.4. Step 4: Selected Images' Brightness and Contrast Assessment*

As part of the selection process, we also assessed the brightness and the root mean square (RMS) contrast values for the 160 images [37,38]. For each image, we calculated the image brightness by averaging the images' RGB pixel values and the RMS contrast, which is the standard deviation of the brightness values as follows:

$$B_{avg} = \frac{1}{N}\sum_{k=1}^{N} L_k \tag{1}$$

$$C_{rms} = \sqrt{\frac{1}{N}\sum_{k=1}^{N}\left(L_k - B_{avg}\right)^2} \tag{2}$$

where $N$ is the total number of pixels, $k$ is the pixel index, $L_k$ is the pixel value, $B_{avg}$ is the average brightness value of an image, and $C_{rms}$ is the RMS contrast of an image. We then estimated the mean and std. dev of these values in the entire set of 160 images and in presumed residents and tourists category sets of 80 images each to check if they lay within three std. dev from the mean. Barring the RMS contrast of one image belonging to the presumed residents category, all the images' brightness and contrast values satisfied this criterion. After reducing the contrast value of this image by 12%, all values were within three standard deviations from the mean for the entire set and separate sets of presumed residents and tourists categories.

**4. User Notes**

We used the social media platform Flickr to create an urban image stimulus set of the city of Lisbon. This image set is a unique and valuable resource that can help research in a variety fields, such as understanding social media use and urban planning. The dataset comprises 160 geotagged urban space images, and these images can be further categorized into 80 images of presumed residents and tourists. In addition to latitude and longitude information, we created variables/attributes associated with these images, such as cell image density, cell green index, and normalized popularity. We conducted robust correlations among these output variables to ensure their orthogonality in both the overall dataset and the category subsets. Further, the brightness and contrast values for the cumulative and within-category image set are within three standard deviations from the mean.

The presented urban image stimulus set is helpful for studying and comprehending different facets of urban settings, such as people's perceptions, preferences, and behav-

iors. For example, the stimulus set can be effectively used to assess urban perception by seeking responses to factors such as valence, arousal, feelings of safety, etc. They can also be combined with personality and mental health questionnaires. By leveraging this combined methodology, researchers can better understand the complex dynamics between physical spaces, individual characteristics, and digital behavior linked to urban spaces. For instance, researchers can use the stimulus set to assess how much digital popularity (measured by normalized popularity) overlaps with real-life location popularity (based on people's preferences and perceptions of the urban environment). This image set can also be employed in neuroimaging experiments using techniques such as functional magnetic resonance imaging (fMRI) and electroencephalography (EEG) [39]. This approach enables researchers to uncover the underlying neural mechanisms and cognitive processes involved in perceiving urban spaces in Lisbon. For example, Chang et al. [40] used fMRI to shed light on the neural mechanisms contributing to the positive mental health outcomes associated with green spaces. In another study, Olszewska-Guizzo et al. [41] conducted a study using EEG to examine the linkage between urban green spaces and mental health outcomes. Our stimulus set includes a metric for measuring greenness called the cell green index and other measurements that offer possibilities for investigating similar and varied neuroimaging-based research questions.

Finally, researchers can combine data from neuroimaging experiments conducted using the urban image stimulus set with machine learning algorithms and explore promising ways to identify patterns and predict future trends across various domains, including urban planning, social media, and interventions in clinical populations. However, it is important to acknowledge that while our set of stimuli covers a significant expanse of 158.43 km$^2$ within Lisbon, other researchers can develop a similar set of stimuli encompassing a diverse range of geographical regions for their respective studies. Further, it should be noted that we divided the Lisbon region into cells of $100 \times 100$ m, and all images within each cell were assigned the same cell green index and cell image density. Therefore, while a particular cell may have a high level of greenery overall, certain images within that cell may contain less or no green coverage.

It is important to note that the urban stimulus set is limited to a specific time frame and is therefore static in nature. Future research holds the potential to enhance our understanding of urban dynamics comprehensively by integrating dynamic data sources and leveraging the capabilities of both community-based geoportals and Spatial Data Infrastructures (SDIs). These combined contributions form a holistic approach that benefits both localized and broader perspectives on the urban environment [42,43]. In addition, it is important to encourage future research in developing stimulus sets specifically tailored to rural areas. This will enable valuable comparisons with urban settings and help uncover unique challenges and opportunities. By doing so, targeted interventions can be created to improve rural life and foster sustainable development. Despite the above points, our urban image stimulus set can be very useful to researchers and contribute to developing innovative solutions. Researchers can tackle complex environmental and social science challenges by combining this stimulus set with self-reported evaluations, neuroimaging methods, and advanced machine learning algorithms. Additionally, the output variables included in this set allow for analysis of the intricate relationship between human behavior, decision-making, social dynamics, and the environment.

**Author Contributions:** A.K. contributed to stimulus set creation, created stimulus selection methodology, wrote the original draft, and revised the manuscript. A.L.R. contributed to stimulus set creation and wrote and edited the manuscript. S.H. contributed to stimulus set creation. D.A.B.-M. contributed to stimulus set creation and edited the manuscript. B.M. conceptualized and supervised stimulus set creation, secured funding, and edited the manuscript. P.M. conceptualized and supervised stimulus set creation, secured funding, and wrote and edited the manuscript. D.M. conceptualized and supervised stimulus set creation, secured funding, and wrote and edited the manuscript. All authors have read and agreed to the published version of the manuscript.

**Funding:** This work is a part of the eMOTIONAL Cities project, which received funding from the European Union's Horizon 2020 research and innovation programme under grant agreement No. 945307.

**Institutional Review Board Statement:** Not applicable.

**Informed Consent Statement:** Not applicable.

**Data Availability Statement:** We have provided this urban image stimulus set and the MATLAB code for the algorithm utilized for stimulus set selection for download on the Open Science Framework (OSF) (https://osf.io/c79w5/, accessed on 23 November 2023).

**Conflicts of Interest:** The authors declare no conflict of interest.

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
