# Peer review of "An Urban Image Stimulus Set Generated from Social Media"

_data_

Round 1
Reviewer 1 Report
Comments and Suggestions for Authors
In general, good research has been done to prepare a geo-tagged data set and calculate specific indices such as Cell Image Density, Cell Green Index, etc. as an example for the city of Lisbon. However, there are also important comments that I think should be addressed.
- Unfortunately, from reading the abstract, I did not understand what the main purpose of the research and its contribution was. It is also necessary to explain which scientific background the subject of the research is related to and what research gap existed in this field that led to the present research.
- The link related to OSF did not open for me, if possible fix the problem.
- Lines 84 to 93 deal with the implementation details in the Introduction section, which I think should be deleted. Instead, given that the concept of sharing geo-tagged data was first introduced by researchers in the neogeography with keywords such as 'volunteered geographic information', 'crowdsource mapping', and 'citizen science', you should state your research position in this area of science by referring to related articles.
- Line 111: I was interested to see the dataset but unfortunately the link provided (https://osf.io/c79w5/?view_only=0e227f28d8504cc9b677f25d8f595f88) did not work.
- The conducted research deals with preparing a data set from social media and enriching it with special descriptions such as cell green index and cell image density. Due to the dynamic nature of social media and real-world transformation, the number of visitors, the amount of vegetation density, the number of photos, etc., change in a very short period of time, and the value of these indicators quickly disappears, so that they can no longer help the researchers. Therefore, it is strongly recommended to at least discuss (Maybe in a separate Discussion Section) the necessity of developing a community-based geoportal for the up-to-date generation of geo-tagged datasets and the related specialized indices for the purpose of this research and compare this idea with similar works such as Vahidnia and Vahidi (2021) and De Longueville (2010).
Reviewer 2 Report
Comments and Suggestions for Authors
The article seems valuable, especially in its technical part that concerns Data. However, in my opinion, the weakest part of the article is the section: 1. Summary, which is not very convincing for me. Detailed comments below:
1. In line 37, the authors mentioned their chosen social media platforms, why? Why did the authors choose: Facebook, Instagram, Twitter and TikTok? First of all – the Twitter platform has changed its name to the X platform – this should be noted in the text. Why did the authors mention these specific online platforms? Is it related to their reach and popularity? If so, I suggest supporting it with relevant literature. Please comment.
1a) The name Twitter appears several times in the article. In my opinion, the article should emphasize that currently the website is called "X", e.g. by adding a note in brackets: (currently website "X").
2. On lines 41-42 the authors wrote: "Moreover, certain platforms also allow users to add geographical location tags to their posted content, to indicate the users' location and/or where the content was captured." – Which platforms are these? Please provide specific examples, similar to those mentioned in line 37. Does this only apply to Flickr?
2a) Moreover, in lines 50-51 the authors wrote: "Geotagged media plotted on maps allows for visualizing the distribution and concentration of these data across different locations, giving insights into preferences (most sought-after locations), popular activities, interests, and emerging trends) – what media? Any specific ones? Please comment.
3) In section 1. Summary, the authors focused exclusively on global urbanization, cities and urban agglomerations (quote: 57% of the world's population currently living in urban areas). The authors cited the results of research carried out in cities. But what about the remaining 43% of the population? Are there any results from studies carried out in rural areas? It is worth emphasizing here that it is the rural areas and agriculture that keep cities alive. Please comment.
4) Lines 79-83 – “In sum, geotagged data can play an important role in urban planning and development initiatives, enabling planners to make data-driven decisions to improve urban livability and sustainability and help policymakers to derive policy recommendations tailored to local needs." What about rural areas? Doesn't this apply to rural, marginalized, remote, mountainous, etc. areas? Is big data just a city? Please comment.
5) In my opinion, the presented method is "static". What I mean is that research results are a "snapshot of a specific condition" that occurred during the period from which the data comes. In my opinion, this is a certain weakness of these studies. Today we are interested in real-time results, i.e. results obtained on an ongoing basis. Please comment.
6) I felt that the conclusions presented in section 4. User Notes went too far. The authors described their research in only superlatives. In my opinion, the disadvantages, limitations and weaknesses of the presented research should also be described. Please comment.
7) The illustrations are of poor quality (blurred rasters). Perhaps formatting the article by the editors will improve the situation.
Round 2
Reviewer 1 Report
Comments and Suggestions for Authors
In my opinion, the corrections made make the article acceptable.
Reviewer 2 Report
Comments and Suggestions for Authors
The corrections are satisfactory to me.
1) One minor suggestion remains, namely: the hyperlink in its current form, i.e.
https://osf.io/c79w5/?view_only=0e227f28d8504cc9b677f25d8f595f88,
is long and "ugly", unprofessional, completely inconsistent with SEO.
Can't it be replaced with a nice alias, e.g. from Bitly.com? Or maybe the "osf.io" website allows you to create SEO-compliant short links, or a DOI link leading to the data? The hyperlink mentioned above looks very bad. For your consideration.
2) Moreover, next to the hyperlink, to the best of my knowledge, there should be the so-called access date, or maybe it doesn't have to be (?)
